# Recent Advances in Small Molecule Inhibitors for the Treatment of Osteoarthritis

**DOI:** 10.3390/jcm12051986

**Published:** 2023-03-02

**Authors:** Jianjing Lin, Shicheng Jia, Weifei Zhang, Mengyuan Nian, Peng Liu, Li Yang, Jianwei Zuo, Wei Li, Hui Zeng, Xintao Zhang

**Affiliations:** 1Department of Sports Medicine and Rehabilitation, Peking University Shenzhen Hospital, Shenzhen 518036, China; 2Shantou University Medical College, Shantou 515041, China; 3Department of Bone and Joint, Peking University Shenzhen Hospital, Shenzhen 518036, China; 4National & Local Joint Engineering Research Center of Orthopedic Biomaterials, Peking University Shenzhen Hospital, Shenzhen 518036, China; 5Peking University Shenzhen Hospital, Shenzhen 518036, China

**Keywords:** osteoarthritis, small molecule, inhibitors

## Abstract

Osteoarthritis refers to a degenerative disease with joint pain as the main symptom, and it is caused by various factors, including fibrosis, chapping, ulcers, and loss of articular cartilage. Traditional treatments can only delay the progression of osteoarthritis, and patients may need joint replacement eventually. As a class of organic compound molecules weighing less than 1000 daltons, small molecule inhibitors can target proteins as the main components of most drugs clinically. Small molecule inhibitors for osteoarthritis are under constant research. In this regard, by reviewing relevant manuscripts, small molecule inhibitors targeting MMPs, ADAMTS, IL-1, TNF, WNT, NF-κB, and other proteins were reviewed. We summarized these small molecule inhibitors with different targets and discussed disease-modifying osteoarthritis drugs based on them. These small molecule inhibitors have good inhibitory effects on osteoarthritis, and this review will provide a reference for the treatment of osteoarthritis.

## 1. Introduction

Osteoarthritis (OA) refers to a degenerative disease with joint pain as the main symptom, and it is caused by various factors, including fibrosis, chapping, ulcers, and loss of articular cartilage [1]. The etiology is not clear, and its occurrence is closely related to age, obesity, inflammation, trauma, and genetic factors [2]. OA tends to occur in middle-aged and elderly people, with a high incidence, and more than 50% of people over 65 years old are OA patients. The affected areas include joints such as knees, hips, ankles, hands, and spine (cervical spine and lumbar spine) [3]. The treatments of OA include basic treatment, medical treatment, and surgical treatment. Basic treatment generally includes health education, exercise therapy, and physical therapy. Medications are generally non-steroidal anti-inflammatory drugs, analgesics, intra-articular injections, slow-acting drugs to relieve symptoms of OA, and anti-anxiety drugs. However, these treatments can only delay the progression of OA, and patients may eventually need joint replacement [4,5,6], so OA urgently needs new medical treatment. Small molecule inhibitors may be the hope for the treatment of OA in the future.

As a class of organic compound molecules weighing less than 1000 daltons, small molecule inhibitors are the basis of most medications in the study of signal transduction pathways because they can target specific proteins to reduce their activity or interfere with biochemical reactions [7,8,9]. Apixaban, a blood clotting factor X inhibitor, is clinically used for adult patients undergoing elective hip or knee replacement surgery to prevent venous thromboembolism, and it was in second place in global drug sales in 2019, proving the great role of small molecule inhibitors in disease treatment [10,11,12]. The fifth place on the global sales list in 2019 was ibrutinib, which is a highly effective and selective small molecule BTK inhibitor for the treatment of chronic lymphocytic leukemia and small lymphocytic lymphoma [13,14,15]. Small molecule inhibitors for OA are also under constant research. Venkatesan Aranapakam et al. discovered that alpha-sulfonylhydroxamic acids can inhibit the activity of MMP as an effective small molecule inhibitor for the treatment of OA [16]. In addition, there are many other studies on small molecule inhibitors for OA. In this regard, by reviewing relevant manuscripts, small molecule inhibitors targeting matrix metalloproteinase (MMP), ADAMTS, IL-1, TNF, WNT, NF-κB, and other proteins, with good inhibitory effects on OA, were reviewed. This review could provide a reference for the treatment of OA (Shown as Figure 1 [17,18,19,20,21,22]).

## 2. Main Molecular Mechanisms of OA

The main molecular mechanisms of OA are caused by the proliferation of cartilage cells and the progressive loss of extracellular matrix (ECM) proteins. Though the cartilage surface remains intact at the beginning, the first changes in the molecular composition of the tissue and ECM are characterized by increased synthesis of ECM proteins such as type II collagen and aggrecan, with chondrocyte proliferation. Along with the development of OA, the synthesis of degrading proteases increases, type II collagen is degraded, and its degraded fragments stimulate proteins associated with the catabolic state, such as MMPs and ADAMTSs [23].

IL-1β, produced by chondrocytes, osteoblasts, synovial cells, and leukocytes, is considered the most effective cytokine in degrading cartilage in the pathogenesis of OA. IL-1β binds to IL-1R in synoviocytes, which activates transcription factors through NF-κB and MAPK signaling pathways, regulates the inflammatory response, and leads synoviocytes to produce inflammatory mediators such as MMPs, COX-2, PGE2, NO, and other catabolic factors, accelerating cartilage degradation [23,24,25]. Moreover, IL-1β activates downstream gene expression and production of MMP family enzymes such as MMP-1 and MMP-13, and a disintegrant and metalloproteinase with thrombospondin motifs (ADAMTS) and other enzymes mainly break down cartilage cells and extracellular matrix to destroy the cartilage structure [26,27,28,29,30,31]. In addition, IL-1β stimulates the production of reactive oxygen species, such as peroxide and hydroxyl radical formation, which can directly damage articular cartilage during disease progression [23,24,25]. As the most important inflammatory factor that stimulates the inflammatory response of OA, TNF-a can act alone or synergistically with IL-1β, IL-6, and other cytokines to stimulate the cartilage, synovium, and subchondral bone layer-related cells to produce MMP-1, MMP-3, and MMP13 to release cartilage collagen. Specifically, free TNF-α binds to its corresponding receptors and activates transcription factors through the NF-κB signaling pathway, producing an inflammatory response similar to but weaker than that of IL-1β. In addition, the effect of TNF-α can also be mediated through the PI3K/Akt axis, which can increase the expression of cadherin-11 in fibroblast-like synoviocytes, leading to an increased degree of synovitis and cartilage destruction [23,32,33]. Based on the mechanisms, the protein and proteoglycan are gradually lost, and the synthesis of proteoglycan and type II collagen is inhibited, thus leading to the death of chondrocytes and breaking the steady-state balance between cartilage damage and repair [34,35,36,37,38]. IL-6 is another proinflammatory factor that is abnormally elevated in synovial fluid or serum of OA patients. It is a key cytokine that can cause a high level of MMPs and changes in the subchondral bone layer, including promoting synovial tissue inflammation, increasing cartilage permeability, accelerating osteoclast formation, and causing cartilage absorption, degradation, and destruction [23,32,33]. In addition, although IL-6 itself has no direct effect on the synthesis of proteases, PGE, or matrix proteins, it may produce synergistic effects with factors such as TNF [39]. At the same time, Liao et al. [40] proposed that simultaneous inhibition of SATA3 and ERK on the regulation of IL-6 downstream signaling could improve the pain and cartilage decomposition symptoms of OA patients. The activation of the IL-6 signaling pathway under the regulation of the MAPK signaling pathway in synoviocytes and chondrocytes will further start the signaling cascade, leading to the release of inflammatory mediator IL-6 and articular cartilage damage [41].

All in all, it is thought that the molecular mechanism of OA is caused by the breakdown of cartilage anabolism and catabolism, and the catabolism is mainly IL-1β and TNF that can activate MMP family genes and ADAMTS. Therefore, the targets of small molecule inhibition are mainly these four genes. Of course, there are also small molecule inhibitors that target other genes, and they are summarized later in this review.

## 3. Different Targeted Small Molecule Inhibitors for OA

### 3.1. Small Molecule Inhibitors Targeting the MMP Gene Family for OA

MMPs are related to many diseases, including OA, and the specific mechanism may cause MMPs to degrade cartilage. Therefore, synthetic small molecule inhibitors targeting MMPs should be a promising treatment for OA [16,42,43,44,45].

A Yamada et al. developed a small molecule inhibitor ONO-4817 that targets MMPs. Furthermore, ONO-4817, a hydroxamic acid-based nonpeptide compound, can broadly inhibit MMPs other than MMP-1 and MMP-7 through injecting with LPS into the knee joints of 42 guinea pigs. The proteoglycan in the synovial fluid of the knee joints decreased with the increase in the concentration of ONO-4817, which indicates that ONO-4817 may have the possibility of treating MMP-related diseases, including OA [17]. H Yamada and others have also developed a small molecule inhibitor CPA-926 that can be orally targeted to MMPs. On the other hand, they induced OA in the knee joints of 15 Japanese adult white rabbits through lateral meniscus resection, and 10 rabbits were orally administered CPA-926 at 200 or 400 mg/kg/day for 14 days. Compared with the control group, the rabbits treated with CPA-926 at a dose of 400 mg/kg showed a reduction in the size of the tibial plateau erosion zone, and CPA-926 can also inhibit the expression of MMP-1 and MMP-3. These results indicated that the prophylactic administration of CPA-926 seems to provide some protection against cartilage destruction in the short-term rabbit experimental OA model [46]. In addition, some small molecule inhibitors that target MMP have made some progress in clinical trials. According to E J Lewis et al., Ro32-3555, an oral collagenase inhibitor, has been found to inhibit cartilage degradation in vivo and in vitro as a potential treatment for OA [42,47]. The above-mentioned experimental studies have made great contributions to small molecule inhibitors targeting MMPs. Of course, there are some small molecule inhibitors of OA targeting MMPs that have been developed through design and computer prediction. Venkatesan Aranapakam et al. found that alpha-sulfonylhydroxamic acids can be used as a new type of oral MMP inhibitor for the treatment of OA [16]. Janelle L Lauer-Fields and others have also obtained some effective small molecule inhibitors through high-throughput screening [48].

It can be seen from the above that there is sufficient clinical research on small molecule inhibitors targeting MMPs. However, the preclinical data are not enough to allow these inhibitors to be commoditized. In addition, there are many small molecule inhibitors of OA targeting MMPs, which are expected to become the core therapies for OA in the future [49]. Our summary is listed in Table 1 [16,17,46,47,48,49].

### 3.2. Small Molecule Inhibitors Targeting ADAMTS for OA

According to previous studies, the ADAMTS gene family plays a very important role in OA, in which ADAMTS4 and ADAMTS5 are two key factors, so it becomes natural for small molecule inhibitors to target ADAMTS [50,51,52,53,54].

According to Hongfeng Deng et al., propylbenzenesulfonamide (8) can inhibit the occurrence and development of OA by targeting ADAMTS-5, which could provide the first small molecular basis for targeting ADAMTS-5 to inhibit the occurrence and development of OA [55]. Shinya Sogame and others also proved that, as a potentially effective small molecule inhibitor of OA, benzimidazole can target ADAMTS-5 to inhibit the activity of ADAMTS-5 [56]. Anne Sofie Siebuhr et al. discovered that a novel anti-ADAMTS-5 inhibiting nanobody M6495 can effectively inhibit the activity of ADAMTS-5 in cells, which could provide a basis for the treatment of OA [18].

The above studies confirmed that the small molecule inhibitors targeting ADAMTS showed gratifying results in experimental OA, and we look forward to its further research for clinical use. Our summary is listed in Table 2 [18,55,56].

### 3.3. Small Molecule Inhibitors Targeting IL-1 for OA

IL-1 has a full range of effects on OA, and it plays a role in changing the normal chondrocyte structure and function, promoting chondrocyte apoptosis, degrading the chondrocyte matrix, participating in slippage membranous inflammatory disease, and affecting bone metabolism and other aspects. Therefore, the development of small molecule inhibitors targeting IL-1 will be a very suitable choice [57,58,59,60,61,62].

Zuqiang Wang et al. proposed that anemonin can target IL-1 and inhibit the progression of OA at the cellular and animal levels, and it is an effective small molecule for protecting against OA [19]. Shuang Liang et al. also put forward that Necrostatin-1 can inhibit the activity of IL-1. Both in vivo and in vitro tests showed significant inhibition of the progression of OA, so Nec-1 may be a new drug candidate for the treatment of OA [63]. Jie Shen et al. found that an FDA-approved small molecule Abat inhibitor, vigabatrin, can inhibit the expression of IL-1, inhibit the progression of OA, and contribute new methods to the treatment of OA [64]. AL Akeson et al. revealed that AF12198 can directly target IL-1 and inhibit IL-1-related diseases, including OA [65].

From the above, it can be known that small molecule inhibitors targeting IL-1 are a promising treatment for OA. Small molecule inhibitors related to IL-1 are divided into those directly combined with IL-1 and those combined with IL-1R. The different classes present different mechanisms, routine administrations, and therapeutic effects. More research needs to be performed to identify whether the differences exist. There are other studies showing that targeting IL-1 is a potential treatment for OA in the future [66]. Our summary is listed in Table 3 [19,63,64,65,66].

### 3.4. Small Molecule Inhibitors Targeting TNF for OA

Tumor necrosis factor (TNF) is derived from macrophages, fibroblasts, chondrocytes, etc., and it is an important mediator in the degradation of the cartilage matrix. TNF can stimulate synovial cells to produce PGE and induces peroxidation in chondrocytes, increasing the destruction of cartilage. Therefore, targeting TNF to design small molecule inhibitors to treat OA is also a reasonable strategy [67,68,69,70,71].

Feng-Cheng Liu et al. isolated chondrocytes from the joints of pig joints or OA patients undergoing total knee replacement surgery, established an OA model, and screened Cf-02, a small molecule inhibitor that inhibits inflammation and cartilage damage [20]. Cecilie F Kjelgaard-Petersen et al. illustrated that tofacitinib and TPCA-1 can inhibit the occurrence and development of OA by inhibiting the expression of TNF [72]. Although these compounds showed an inhibitory effect on cartilage degradation through targeting TNF, serum biomarkers indicate that these processes may be driven by different pathways.

TNF plays an important role in OA, so it is hoped that more targeted TNF small molecule inhibitors will be used in the clinical treatment of OA in the future. Our summary is listed in Table 4 [20,72].

### 3.5. Small Molecule Inhibitors Targeting Other Proteins for OA

In the occurrence and development of OA, in addition to the above four proteins, many other proteins also play a significant role that cannot be ignored. Therefore, the development of small molecule inhibitors targeting other proteins has also become a good choice [73,74,75,76,77,78,79].

#### 3.5.1. Wnt

The Wnt signaling pathway is a type of signaling pathway that is mainly mediated by β-catenin and involves multiple proteins. The Wnt/β-catenin signaling pathway can regulate the development of OA, so targeting WNT-related proteins may be beneficial for OA [80,81,82,83,84,85,86,87,88].

Y Yazici et al. recruited 61 subjects and divided the patients into four groups. The patients were injected with 0.03, 0.07, or 0.23 mg of SM04690 or placebo. The authors found that SM04690 was safe and well tolerated and was effective in improving OA pain. Moreover, the trend of inhibiting the progress of OA is gratifying, and the specific mechanism is the targeting of the Wnt signaling pathway to inhibit its activity and inhibit the occurrence and development of OA [22]. V Deshmukh et al. discovered the direct mechanism of SM04690 inhibiting the WNT signaling pathway through biochemical experiments and Western blotting: SM04690 inhibits the activity of the WNT signaling pathway by inhibiting CLK2 and DYRK1A [89].

According to Caressa Lietman et al., XAV-939 is a small molecule inhibitor of Wnt/β-catenin signaling, and it improves the severity of OA related to cartilage degradation and synovitis reduction in the body. Therefore, XAV-939 is a small molecule inhibitor of Wnt/β-catenin signaling and a promising small molecule inhibitor targeting the WNT signaling pathway. It is hoped that it will become a therapeutic drug in the clinic in the future [90]. Akira Takamatsu et al. concluded that verapamil, a hypertensive drug widely used clinically, also has the effect of inhibiting OA. Verapamil inhibits the Wnt signaling pathway by inhibiting the nuclear localization of β-catenin, and in vivo and in vitro experiments revealed the suppression of the progression of OA [91].

SM04690, a small molecule drug targeting the WNT signaling pathway, has entered clinical trials and has become a promising drug for small molecule inhibitors for the treatment of OA in the future. It is hoped that it can become a real drug for the treatment of OA in the future [92].

#### 3.5.2. NF-κB

NF-κB is a transcription factor that induces immune response, inflammation symptomatic response, cell differentiation, and survival of normal and malignant cells. It plays an important role in OA, and the inflammatory process and mechanical load of OA can activate the NF-κB pathway and speed the catabolism of chondrocytes [93,94,95,96,97].

Pei-Wei Weng et al. revealed that SC75741 can inhibit the occurrence of OA by inhibiting the NF-κB signaling pathway, which could prove the therapeutic potential of SC75741, as a small molecule inhibitor of OA progression [21]. Zhu Xu et al. demonstrated that nepetin can inhibit the NF-κB signaling pathway with protective and therapeutic effects on mouse OA models, which suggested that nepetin is a new potential treatment option in OA. Therefore, it can be said that small molecule inhibitors targeting NF-κB have great potential in the treatment of OA [98].

#### 3.5.3. Others

Of course, apart from MMPs, ADAMTS, IL-1, TNF, WNT, and NF-κB, there are many small molecule inhibitors that can inhibit the occurrence and development of OA. Fumiko Yano et al. proposed that TD-198946 can target Runx1 to inhibit OA, which indicated that TD-198946 is a new class of drug candidates and targeting Runx1 will provide a promising new method for the development of disease-relief drugs for OA [99].

Anastasios Chanalaris et al. discovered that Suramin can target metalloproteinase 3 to inhibit OA and cartilage degradation and can be regarded as a promising small molecule inhibitor for the treatment of OA [100].

In fact, there are many small molecule inhibitors that target other proteins to inhibit OA, as shown in Table 5 [99,101,102,103,104,105,106,107,108,109].

## 4. Small Molecule Inhibitors as DMOADs

Disease-modified OA drugs (DMOADs) are a class of drugs that delay or even reverse the progression of an OA by targeting its underlying cause. The DMOADs undergoing clinical trials are summarized in Table 6. With advances in understanding and research on pathophysiology and small molecule inhibitors, the ideal DMOAD which targets pathological progressions such as activation of matrix-degrading enzymes, disturbance of Wnt pathway, remodeling of subchondral bone, and synovial inflammation; prevents the loss of cartilage; and potentially regenerates cartilage has become more significant [68,110].

### 4.1. DMOADs Based on Protease Inhibitors

Data from mouse models have emphasized the effect on ADAMTS5 and MMP13 in OA caused by cartilage injury [111,112,113,114], which proved that inhibiting the activity of matrix-degrading enzymes is a strategy to limit cartilage damage. DMOADs are developed in this field. GLPG1972, one of the small molecule inhibitors of ADAMTS5, has been demonstrated to have an acceptable effect in the protection of cartilage in an OA model [115,116]. Currently, a phase I clinical trial of GLPG1972/S201086 has demonstrated that it has a good safety profile and excellent targeting features by finding a decrease in the serum concentration of the aggrecan ARGS neoepitope [117]. Another treatment being investigated is anti-ADAMTS5 antibodies. M6495, known as a single-domain antibody fragment containing the properties of heavy-chain-only antibodies for ADAMTS5, could be used to inhibit aggrecan turnover to improve cartilage injury [118,119]. A phase I study in 2019 indicated that it has an acceptable safety profile and dose-dependent signs of target engagement [120].

Another strategy to limit cartilage injury is inhibiting the activity of matrix-degrading enzymes, although its safety may be less than that of inhibiting aggrecanase because the latter can be cleared by a non-proteolytic mechanism, which prevents accumulation of newly formed aggrecan in cartilage [115]. However, researchers found MMP inhibitors, such as PG-116800, would cause reversible musculoskeletal adverse events in OA patients in an exposure-dependent and dose-dependent way [121]. Thus, inhibiting ADAMTS5 could represent a more promising strategy for the development of DMOADs than MMP inhibitors.

### 4.2. DMOADs Based on Anti-Senility

Cellular senescence can cause immune system disorders, mitochondrial dysfunction, oxidative stress disorders, and abnormal secretion of cytokines [122]. In addition, the number of chondrocytes and synovial fibroblasts in the joint is closely related to age, and joint injury can accelerate the senescence of chondrocytes and stimulate cartilage degradation, which indicates that aging is an important mechanism causing the development of OA [123,124,125]. A variety of related DMOADs are currently being developed.

Several randomized, placebo-controlled clinical trials are ongoing for UBX0101, a p53/MDM2 interaction inhibitor that reduces cytokines and chemokines secreted by senescent cells in OA joints [125,126]. However, almost all previous publications did not carefully control the number of aging markers or the selection of patients, most of whom had clinically advanced disease [127]. Therefore, developing DMOADs remains challenging. For instance, whether disease modification translates into clinical benefit is uncertain as structural changes and function in OA are relatively discordant. In addition, related clinical trials require very long follow-up periods due to the long disease course. More clinical trials currently underway are expected to improve OA management in the coming years. We review more clinical trials in the next section. The summary is listed as Table 6 [22,128,129,130,131,132,133,134].

**Table 6 jcm-12-01986-t006:** The summary of some developed DMOADs.

Drug	Clinical Trial Status	Mechanism of Action	Trial Registration	Reference
BMP7	A phase 1 safety and tolerability study of BMP-7 in symptomatic knee osteoarthritis was conducted in 2010 and demonstrated that there was no dose-limiting toxicity identified.	Upregulates transcription of osteogenic genes	NCT00456157	[128]
Invossa-K	As of January 2022, a phase 3 clinical trial has resumed in the US.	Gene therapy	NCT03291470	-
KA34	A phase 1 study started in May 2018 to evaluate the safety of KA34 in humans.	Induces MSCs to differentiate into chondrocytes	NCT03133676	[129]
SM04690	A phase 2 study was completed, and it demonstrated improvements in pain, function, and joint space width.	Wnt pathway inhibitor	NCT02536833	[22,130]
Sprifermin	A 5-year, dose-finding, multicenter randomized clinical trial conducted at 10 sites demonstrated that intra-articular administration of 100 μg of sprifermin every 6 or 12 months results in significant improvement in cartilage thickness.	Promotes chondrogenesis through fibroblast growth factor receptor FGFR3	NCT01919164	[131]
TPX-100	A one-year study of 104 patients showed that TPX-100 treatment was associated with significant and sustained improvement in the preservation of cartilage thickness and reduced pathological changes in cartilage.	Induces articular cartilage formation and reduces pathological shape change of joint bones	NCT01925261	[132,133,134]

### 4.3. Drug Delivery of DMOADs Based on Small Molecule Inhibitors

As described in this section, there is an urgent need to develop DMOADs to effectively treat OA. The route of administration of a DMOAD largely determines its efficacy and safety, especially determining the therapeutic effect of the drug.

Oral administration and intravenous administration are the most widely used methods of treatment for early-stage OA. Traditional drugs, such as NSAIDs, are administered orally, and their active ingredients pass through the circulatory system to reach the joints. However, because cartilage is nonvascular, only a small amount of the drug can reach the designated site. Currently, some oral small molecule inhibitors, such as ASP7962, NEO6860, and MIV-711, are in or have completed certain stages of clinical trials. Although ASP7962 has not shown obvious superiority over traditional NSAIDs [135], MIV-711 has been confirmed to effectively delay cartilage lesions, and both it and NEO6860 have shown good safety [136,137]. However, limited by the follow-up time, the specific risk–benefit ratio and possible potential adverse reactions have not been clearly reported [136,138], and no detailed study has pointed out the specific differences in the bioavailability of MIV-711 administered orally compared with other administration methods. These factors affect its further development and promotion.

Previous studies have shown that topical/transdermal administration in the form of emulsions or patches can significantly reduce toxicity and irritation, but fewer drugs can cross the skin barrier [139]. To overcome the problem of low skin permeability, the microneedle, a new mode of drug delivery that can penetrate the dermis, has been proposed [140]. However, the rapid decomposition of injected drugs is an urgent problem to be solved.

It is generally believed that intra-articular injection of drugs delivered to synovial joints has the advantages of higher intra-articular drug concentration and fewer side effects compared with traditional oral or local drug delivery, and it has become the main direction for future research. GZ389988A and CNTX-4875 are two types of small molecule drugs that have completed phase II clinical trials, and they are administered intra-articularly. GZ389988A has been confirmed to reduce pain in patients with OA and has considerable safety [141]. CNTX-4875 can significantly improve the pain caused by knee OA, and this drug effect is highly dose-dependent [142]. However, IA injection of DMOADs is limited based on the poor permeability of drugs within cartilage and the rapid clearance by synovial capillaries and lymphatic vessels of the joint [143]. Therefore, a variety of drug delivery systems for IA injection, such as nanoparticles, liposomes, and gels, have been developed to prolong the bioavailability and circulation time to improve the efficacy of drugs. Agarwal et al. [144] synthesized self-assembled poly (2-hydroxyethyl methacrylate)-pyridine nanoparticles with IL-1Ra on the surface to achieve a therapeutic effect on OA by antagonizing IL-1. The nanocapsules have good biocompatibility and stability and improve the efficiency of drug utilization. Using gels supplemented with synovial cells or cartilage-targeting peptides has good biocompatibility, biodegradability, low toxicity and cytotoxicity, good lubricity, and long joint retention time when loaded with moderate molecular weight small molecule inhibitors, and it can better target the synovium or cartilage [145]. The delivery of small molecule inhibitors using synthetic materials such as nanoparticles, liposomes, and gels as drug carriers should be considered as a new research focus.

## 5. The Undergoing and Completed Clinical Trials of Small Molecule Inhibitors for OA

The development of small molecule inhibitors for the treatment of OA has focused on (1) improving joint structure, (2) slowing disease progression, (3) improving joint pain or symptoms, and (4) improving the prognosis and joint function of patients. According to the pathological characteristics of cell senescence and cartilage degeneration in osteoarthritis, as well as the treatment ideas of inflammation control, subchondral bone remodeling, and peripheral pain control, a variety of potential therapeutic methods are expected to be used in the treatment of osteoarthritis. Therefore, a number of human, double-blind, randomized controlled trials have been conducted to verify the safety and efficacy of these small molecule inhibitors. According to related clinical data, we summarize the clinical trials of the main potential OA drugs in Table 7.

As mentioned before, cartilage is a significant therapeutic target in OA, even though it remains uncertain whether long-term symptoms can be relieved by targeting cartilage. Protease inhibitors and senolytic therapies are considered as promising small molecule inhibitors for OA. Although MMP and ADAMTS inhibitors are common conventional targets, the results of oral medicine of ADAMTS-4 and ADAMTS-5 undergoing human clinical trials have not yet been published [18]. In addition to the previously mentioned GLPG1972, M6495, and PG-116800, lorecivivint is a new promising drug. It regulates the Wnt signaling pathway by inhibiting CDC-like kinase 2 and dual-specificity tyrosine phosphorylation-regulated kinase 1 A [89,146]. A multicenter, randomized, double-blind, placebo-controlled study has been completed and demonstrated that this drug can improve joint function, relieve pain, and attenuate OA damage [89]. Sprifermin, known as a recombinant human fibroblast growth factor 18, binds to activated fibroblast growth factor receptor 3 to promote cartilage repair [147]. A randomized, placebo-controlled phase II clinical trial showed sprifermin increases cartilage thickness and substantially reduces cartilage loss. In another dose-related clinical trial, 100 µg sprifermin was found to protect the lateral femorotibial compartment and cartilage thickness [148]. In addition, using TGF-β type I receptor inhibitors to regulate the level of TGF-β is becoming a new strategy because overexpression of TGF-β1 causes chondrocyte apoptosis and cartilage degeneration. A phase II randomized study showed that allogeneic chondrocytes expressing TGF-β1 relieve pain and improve physical function.

Inflammatory cytokines were detected in both the synovial fluid of OA and the serum of OA patients, which indicated that inflammation is important for OA. As IL-1 is an important small molecule affecting the occurrence and development of OA, if the level of IL-1 can be antagonized or downregulated, it will have a significant impact on the treatment of OA. Despite the widespread presence and pathological effects of IL-1 demonstrated in cell and animal experiments, multiple double-blind, randomized clinical trials of the IL-1 inhibitor lutikizumab have found no significant effect on slowing OA progression or relieving pain compared with placebo [31,59]. Whether a subgroup of OA patients will show improvement in joint pain or benefit from protective effects on human OA joints due to IL-1 inhibition remains an open question. The same problem arises with TNF inhibitors. Randomized, double-blind, placebo-controlled trials for infliximab, adalimumab, and etanercept have not provided evidence to support the use of these OA treatments [149,150]. Therefore, more evidence is required to identify the clear effect of anti-inflammatory drugs in OA treatment.

## 6. Challenges and Future of Small Molecule Inhibitors for OA

The mechanism of pain caused by OA is complex, and it is necessary to consider whether this complex mechanism is the cause of the poor efficacy of traditional analgesic drugs. Research on complex pain mechanisms will also help to further address the issue of analgesic drug development for OA. Multiple pathophysiological processes may be involved in the onset and progression of OA, and in some patients, these pathophysiological processes may interact with each other, thereby amplifying or reducing the pharmacological effect of a drug. We still lack reliable biomarkers from clinical data as evidence to support this hypothesis and identify this group of patients. In addition, the long duration and slow disease progression of OA require long follow-up periods, significantly slowing down the advancement of clinical trials. Therefore, while developing drugs, certain studies are needed to determine the factors that predict OA progression so that drug development targets can be more precisely identified and targeted for clinical trials.

Furthermore, osteoarthritis affects a variety of joints and has a high degree of variability in its clinical and imaging manifestations. Various phenotypes have been proposed to explain this abnormal heterogeneity based on the presence of risk factors. The relevance of these phenotypes is controversial. More than one risk factor may be present in an individual, and several risk factors may share a common pathogenic mechanism [151]. This suggests the need for a more detailed exploration of different risk factors in the development and study of small molecule inhibitors.

Animal experiments and clinical trials have highlighted new therapeutic targets for a variety of OA conditions, and small molecule inhibitors developed for some of these targets have shown promising results in phase II-III studies. drugs such as KA34, SM04690, Sprifermin, and TPX-100 have demonstrated promising effects in analgesia and chondroprotection. They are thus reliable candidates for DMOADs and offer hope for the future of OA management. In addition, DMOADs have great potential for efficient treatment of OA through rational design of drug delivery systems. We propose the following recommendations for future research: (1) exploring the pathogenesis of OA could provide more effective targets for the development of DMOADs; (2) the long-term chronic toxicity of novel drug release systems should be evaluated to ensure their long-term safety; and (3) combination therapy targeting multiple sites/symptoms may be a more effective IA delivery strategy for OA treatment.

## 7. Conclusions

OA refers to a degenerative disease with joint pain. Traditional treatments can only delay the progression of OA, so patients may need joint replacement eventually. Small molecule inhibitors represent a class of organic compound molecules weighing less than 1000 daltons that can target proteins, and the main components of most drugs used clinically are small molecule inhibitors. Small molecule inhibitors for OA are under constant research. We believe that future DMOADs based on small molecule inhibitors should be combined with various modalities such as biological predictors of OA, prognostic factors, and exercise therapy not only for targeting the endogenous heterogeneity of patients, but also for accurately assessing the clinical heterogeneity of patients. In conclusion, drugs and treatment options developed based on small molecule inhibitors are emerging in the field of OA, and the next decade may see significant changes in the management of OA that will hopefully help to reduce the enormous burden of this very prevalent disease.

## Figures and Tables

**Figure 1 jcm-12-01986-f001:**
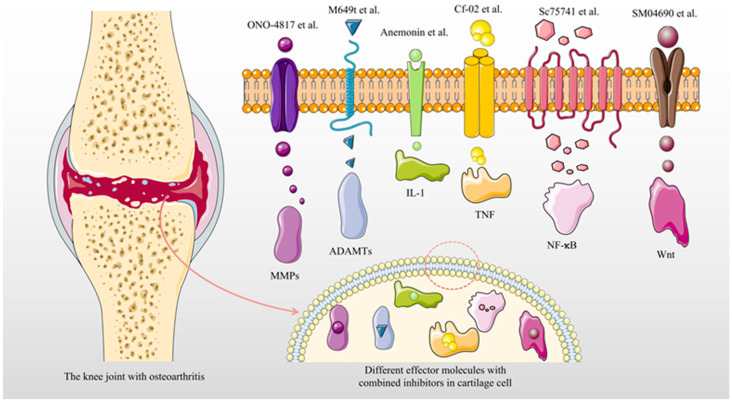
Small molecule inhibitors in osteoarthritis combined with their targeted proteins or genes [17,18,19,20,21,22].

**Table 1 jcm-12-01986-t001:** Small molecule inhibitors targeting the MMP gene family for OA.

Inhibitors	Compound Type	Target	Reference
a-Sulfonylhydroxamic Acids	Sulfonylhydroxamic Acids	MMPs	[16]
Cm-02/Ck-02	oxazine	MMPs	[49]
CPA-926	dihydroxycoumarin	MMP-1 and MMP-3	[46]
ONO-4817	hydroxamic acid	MMPs except MMP-1 and MMP-7	[17]
pyrimidine-4,6-dicarboxylic acid and similar compounds	dicarboxylic acid and similar compounds	MMPs	[48]
Ro32-3555	piperidinobutyrooxamic acid	MMPs	[47]

**Table 2 jcm-12-01986-t002:** Small molecule inhibitors targeting ADAMTS for OA.

Inhibitors	Compound Type	Target	Reference
Propylbenzenesulfonamide (8)	Propylbenzenesulfonamide	ADAMTS-5	[55]
benzimidazole	benzimidazole	ADAMTS-5	[56]
M6495	-	ADAMTS-5	[18]

**Table 3 jcm-12-01986-t003:** Small molecule inhibitors targeting IL-1 for OA.

Inhibitors	Compound Type	Target	Reference
Anemonin	Anemonin	IL-1	[19]
Necrostatin-1	Necrostatin	IL-1	[63]
vigabatrin	vigabatrin	IL-1	[64]
AF12198	carboxylic acid	IL-1	[65]
TAK-242	cyclohexene	IL-1	[66]

**Table 4 jcm-12-01986-t004:** Small molecule inhibitors targeting TNF for OA.

Inhibitors	Compound Type	Target	Reference
Cf-02	Flavonoids	TNF	[20]
Tofacitinib/TPCA-1	-	TNF	[72]

**Table 5 jcm-12-01986-t005:** Small molecule inhibitors targeting other proteins.

Inhibitors	Compound Type	Target	Reference
TD-198946	DMOAD	Runx1	[99]
NSC117079	-	Phlpp	[101]
BNTA	-	SOD3	[102]
AT-406	Smac analog	IAP	[103]
AMD3100	Plerixafor	CXCR4	[104]
Resistin	-	PKCα	[105]
Imperatorin	Coumarins	iNOS	[106]
PBA	Phenylbutyric acid	PERK	[107]
Noggin	-	TGF-β1	[108]
MT-SYK-03	-	SYK	[109]

**Table 7 jcm-12-01986-t007:** Main potential small molecule inhibitors in OA clinical trials.

Drug	Type	Action Mechanism	Route of Administration	Current Phase State	Trial Registration
ASP7962	TrkA inhibitor	Selective inhibition of ATP-induced substrate phosphorylation of human TrkA.	Oral	Phase II: Four-week 100 mg BID ASP7962 did not improve pain or physical function in osteoarthritis patients.	NCT02611466
GZ389988A		A small molecule inhibitor of TrkA.	Intra-articular	Phase II: GZ389988A reduced pain, protected physical function, and had a good safety profile.	NCT02845271
CNTX-4875	TRPV1 modulator	A small molecule modulating TRPV1.	Intra-articular	Phase II: CNTX-4975 provided dose-dependent improvement in knee OA-associated pain. Phase III: Undergoing trial.	NCT02558439 NCT03660943 NCT03661996
NEO6860		An antagonist of TRPV1, blocking the activation by capsaicin.	Oral	Phase I: The first-in-human study on NEO6860 indicated an acceptable safety profile. Phase II: Undergoing trial.	NCT02337543 NCT02712957
Fasinumab	NGF inhibitor	IgG4 anti-NGF monoclonal antibody that binds selectively to NGF.	Subcutaneous	Phase IIb/III: Fasinumab improves OA pain and function. The observed benefit-to-risk relationship favors further clinical development to explore the lowest doses of fasinumab in patients with knee or hip OA.	NCT02447276.
MIV-711	Cathepsin K inhibitor	A novel selective cathepsin K inhibitor	Oral	Phase II: MIV-711 was not more effective than placebo for pain, but it significantly reduced bone and cartilage progression with a reassuring safety profile.	NCT02705625, NCT03037489
Tocilizumab	IL-6R inhibitor	An antibody against the IL-6 receptor that decreases the serum IL-6 level.	Intravenous	Phase III: Tocilizumab was no more effective than placebo for pain relief in patients with hand osteoarthritis.	NCT02477059
UBX0101	Senolytic drug	A p53/MDM2 interaction inhibitor that reduces cytokine and chemokine secretion	Intra-articular	Phase II: Undergoing trial.	NCT04129944 NCT04229225

## Data Availability

No new data were created, not applicable.

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
