# Peer review of "Recent Advances in Small Molecule Inhibitors for the Treatment of Osteoarthritis"

_jcm, 2023, doi:10.3390/jcm12051986_

Round 1
Reviewer 1 Report
First of all, I would like to thank the editors and journal who invited me to review this manuscript.
This manuscript is informative and will be of help to OA-related researchers. Overall, it is well-organized and rich in content, and it deserves publication.
I would like to give you an opinion that you can agree to publication if the manuscript is improved in the following three points.
1) I would like you to separate the contents of the clinical trial into a separate major section, summarize the development situation from phases 1 to 4, and describe the effect size and individual mechanism of each pipeline. A review of a specific class of drugs is considered essential for in-depth investigation and discussion of these contents.
2) DMOAD is considered to be the future target of current OA treatment. To date, no drugs in this class have been approved, but there is much to discuss in the subject of this manuscript. It would be good to separate this part into an independent discussion or perspective section and deal with it in detail.
3) In the discussion or outlook section, it is necessary to describe in detail the existing research and the author's outlook on what kind of clinical findings of OA patients the supplemented contents in 1) and 2) above will eventually show. In addition, regarding the unmet medical needs in the field of small molecule inhibitor drugs and the research methods to solve them, I would like you to cite the new pathology research of OA currently being studied and express the author's view in detail.
With this improvement, it is believed that this manuscript will be welcomed by more OA researchers.
Reviewer 2 Report
Lin et al. review the recent studies on small molecule inhibitors in treating osteoarthritis. The review is generally comprehensive but also overlaps to some extent with another review article published in 2022 (PMID: 35960127). One advantage compared to the aforementioned review is that Lin et al. did a good job in incorporating details in this review. However, there are a few points that need to be addressed before the consideration of publication.
1. The section on molecular mechanism is too short and general. This is an important section as it lays down the foundation of molecular mechanism that the small molecule inhibitors can target. This section needs to be expanded greatly.
2. There is a lack of critical thinking and interpretation of relevant studies in this manuscript. The authors kept using a similar sentence template throughout the manuscript "xxx targets xxx, which prevents OA progression...". In other words, the manuscript only listed the key findings without critically judging the findings or interpretations. For example, what is difference in efficacy and potency among these small molecule inhibitors? Are the pharmacokinetics of the inhibitors different? Are they physiologically relevant in humans? These are just examples and many other aspects can be discussed.
3. The authors need a section on delivery mode of these small molecule inhibitors.
4. The authors need to discuss the role of IL6 in OA progression. There is an elegant recent paper talking about IL6-JAK-STAT3 axis in OA development. Liao et al. even proposed using small molecule inhibitors (JAK and ERK inhibitors) to treat OA in mouse model and these inhibitors have a promising effect on OA prevention (PMID: 35881692).
Author Response
Point 1: The section on molecular mechanism is too short and general. This is an important section as it lays down the foundation of molecular mechanism that the small molecule inhibitors can target. This section needs to be expanded greatly.
Response 1: Thank you for your comments which are indeed our shortcomings and we have made targeted improvements to them. In the second part "Molecular mechanism of OA", we further discussed the pathogenesis of OA and briefly describe the effect of MMPs and ADAMSTs on type II collagen during the early formation and progression of OA: "Along with development of OA, the synthesis of degrading proteases increases, type II collagen degraded, and its degraded fragments stimulate proteins associated with the catabolic state, such as MMPs and ADAMTSs". Also, we added a discussion of the pro-inflammatory factors IL-1β, IL-6, and TNF, interpreting their regulation through various signaling pathways such as NF-κB, wnt, and STATA3, and the promotion of OA progression that develops on this basis. For more details about this section, you can see it in the highlighted section of the manuscript.
Point 2: There is a lack of critical thinking and interpretation of relevant studies in this manuscript. The authors kept using a similar sentence template throughout the manuscript "xxx targets xxx, which prevents OA progression...". In other words, the manuscript only listed the key findings without critically judging the findings or interpretations. For example, what is difference in efficacy and potency among these small molecule inhibitors? Are the pharmacokinetics of the inhibitors different? Are they physiologically relevant in humans? These are just examples, and many other aspects can be discussed.
Response 2: Thank you for your comments. We re-evaluated small molecule inhibitors we mentioned in manuscript. We have discussed some of these drugs in more depth. In the section on inhibitors of MMPs, we present the implications of the lack of preclinical data on drugs for their further development and use, and in the section on IL-1, we point out the possible differences in drug efficacy and clinical safety between small molecule inhibitors with two mechanisms, binding to IL-1 itself and binding to the IL-1 receptor. Based on thses, we focused our outlook on the future development of small molecule inhibitors in a new subsection 6. In this subsection, we focus on the differences in the efficacy of small-molecule inhibitors due to various factors such as drug mechanism, different OA progression, and differences in the mode of administration. The relevant specific statements can be seen in the highlighted section of the newly submitted manuscript.
Point 3: The authors need a section on delivery mode of these small molecule inhibitors.
Response 3: Thank you for your valuable comments. In the newly submitted manuscript, we summarized the different drug effects and clinical trial results from the differences in different routes of administration of DMOAD developed on the basis of small molecule inhibitors, based on the different modes of administration of small molecule inhibitors. Three modes are focused on: oral/transvenous administration; subcutaneous administration; and intra-articular injection administration. The different drugs correspond to different modes of administration and clinical trial results, which have been specifically expressed by us in the article. We also summarized the effects of different modes of administration on drug action. For example, for oral drugs: "Oral and intravenous administration is the most widely used methods of treatment for early-stage OA. Traditional drugs, such as NSAIDs, are administered orally and their active ingredients pass through the circulatory system to reach the joints. But because cartilage is nonvascular, only a small amount of the drug can reach the designated site."; for intra-articular injections: "As OA is a local disease, it is generally believed that intra-articular injection of drugs delivered to synovial joints has the advantages of higher intra-articular drug concentration and fewer side effects compared with traditional oral or local drug delivery, which has become the main research direction in the future. "After summarizing the current state of research, we have placed our outlook on this section in the discussion section of Section 6. A summary of some of the DMOADs that were put into clinical trials can also be read in Table 7.
Point 4: The authors need to discuss the role of IL6 in OA progression. There is an elegant recent paper talking about IL6-JAK-STAT3 axis in OA development. Liao et al. even proposed using small molecule inhibitors (JAK and ERK inhibitors) to treat OA in mouse model and these inhibitors have a promising effect on OA prevention (PMID: 35881692).
Response 4: Thank you for your comments for our shortcoming, and recommendation of such a pretty article. We added the discussion of IL-6 into section 2 “Molecular mechanism of OA” and cited the article you recommended. The contents can be seen: “IL-6 is another proinflammatory factor that is abnormally elevated in synovial fluid or serum of OA patients. It is a key cytokine that can cause a high level of MMPs, and changes in the subchondral bone layer, including promoting synovial tissue inflammation, increasing cartilage permeability, accelerating osteoclast formation, and causing cartilage absorption, degradation, and destruction [16, 19, 20]. In addition, although IL-6 itself has no direct effect on the synthesis of proteases, PGE, or matrix proteins, it may produce synergistic effects with factors such as TNF [21]. At the same time, Liao et al. [22] proposed that simultaneous inhibition of SATA3 and ERK on the regulation of IL-6 downstream signaling could improve the symptoms of OA patients from both pain and cartilage decomposition. The activation of IL-6 signaling pathway under the regulation of MAPK signaling pathway in synoviocytes and chondrocytes will further start the signaling cascade, leading to the release of inflammatory mediator IL-6 and articular cartilage damage [23].”
Reviewer 3 Report
1. the title is not attractive. it can be changed to " Osteoarthritis therapeutic progression: Advances in the application of small molecule inhibitors"
2. The written abstract needs substantial revision
3. There is no need to use abbreviations in the abstract
4. Lines 47-49; it be can be changed to "Small-molecule inhibitors are the basis of most medications in the study of signal transduction pathways because they can target specific proteins to reduce their activity or interfere with biochemical reactions."
5. what means of et al. in figure 1?
6. line 86; are authors mean "Different small molecule targets in the treatment of OA"?
7. Rewrite table 1 based on alphabetical or reference order.
8. table 5 rewrites because some items must be transferred to its section.

Round 2
Reviewer 1 Report
The authors have faithfully reflected my views. This has resulted in an improved manuscript that will be of great benefit to researchers in this field, and I know I will be able to cite the article in any follow-up research I do.
Therefore, I agree to the publication of this manuscript.
Author Response
Thank you for your comment.
Reviewer 2 Report
The revised version has addressed all my points and improved significantly. I recommended it for acceptance.
Author Response
Thank you for your comment.